# Sea ice controls net ocean uptake of carbon dioxide by regulating wintertime stratification
Elise S. Droste [1,2,7] ✉, Dorothee C. E. Bakker [1], Hugh J. Venables[3], Elizabeth M. Jones[4], Michael P. Meredith [3], Giorgio Dall'Olmo [5], Mario Hoppema [2], Oliver J. Legge[1], Gareth A. Lee[1] & Bastien Y. Queste [6]

Sea-air exchange of carbon dioxide in the Southern Ocean is strongly seasonal, with ocean uptake in summer, which is partly offset by carbon dioxide outgassing in winter. This seasonal balance can shift due to sea ice conditions, inducing interannual variability in the Southern Ocean carbon sink. A decade (2010–2020) of unique, year-round marine carbonate chemistry observations from the Rothera Time Series (West Antarctic Peninsula) reveals that interannual variability in seawater fugacity of carbon dioxide depends on wintertime processes. Sea ice duration controls ocean stratification, which acts as a gateway to the carbon-rich ocean interior. Consequently, years with persistent sea ice cover and high mean winter stratification absorb, on average, 20% more carbon dioxide than years with less sea ice and weaker stratification in winter. Wintertime marine observations are therefore essential to resolve critical processes and reliably quantify interannual variability of the sea-air carbon dioxide flux in seasonally sea ice-covered regions.

Observation-based estimates of Southern Ocean sea-air carbon dioxide ($CO_2$) fluxes indicate substantial seasonal, interannual, and decadal variability[1–3]. The interannual variability of the Southern Ocean $CO_2$ flux strongly affects that of the global ocean[4], but global ocean biogeochemical models struggle to capture the amplitude of these year-to-year changes in carbon uptake south of 30°S. The amplitude (<0.1 PgC year$^{-1}$) is generally underestimated compared to observations and agreement amongst models is low[5]. These difficulties may be ascribed to poor reproducibility by models of the Southern Ocean's seasonal cycle of sea-air $CO_2$ exchange, indicating an under-representation of complex interactions between physical and biogeochemical processes, and how sea ice mediates these interactions[5,6]. A mechanistic understanding of the seasonal and interannual drivers behind the Southern Ocean's varying sea-air $CO_2$ flux and the role of sea ice is currently hindered by spatial and temporal data scarcity.

Various data products seek to address this challenge by applying sophisticated interpolation techniques to fill observational data gaps[4,7,8]. However, even between these data products, discrepancies exist in the amplitude of the interannual variability of Southern Ocean $CO_2$ flux, ranging between 0.08 and 0.18 PgC year$^{-1}$ [5]. Underlying this, the seasonality of the carbon cycle in data products has high uncertainties due to undersampling in winter. While biogeochemical Argo floats are closing the winter data gap with pH sensor-based measurements[9], float-based estimates of winter $CO_2$ flux have unresolved uncertainties caused by sensor drift, measurement biases[10], and insufficient ship-based in situ observations for evaluation of winter processes.

In winter, the Southern Ocean releases $CO_2$ to the atmosphere as a result of high surface fugacity of $CO_2$ ($fCO_2$). This is driven by mixing with subsurface carbon-rich Circumpolar Deep Water (CDW)[11]. In summer, surface seawater $fCO_2$ decreases due to high biological productivity[11], allowing the ocean to take up $CO_2$ from the atmosphere, often more than the amount it releases in winter. The Southern Ocean is therefore considered a net atmospheric $CO_2$ sink. The strength of sea-air $CO_2$ flux is positively related to wind speed[12], but is attenuated by sea ice cover, which limits direct sea-air gas exchange[13]. This restrictive effect of sea ice prevents a large fraction of winter $CO_2$ outgassing in seasonally sea ice-covered regions[3]. Similarly, it can prevent atmospheric $CO_2$ uptake if seawater $fCO_2$ is low, as has been observed in the Arctic (e.g., ref. 14). Sea ice is therefore key in maintaining a net atmospheric $CO_2$ sink, because changes in its duration are able to shift the seasonal balance between summer $CO_2$ uptake and winter outgassing.

[1]School of Environmental Sciences, University of East Anglia, Norwich, UK. [2]Alfred Wegener Institute Helmholtz Centre for Polar and Marine Research, Bremerhaven, Germany. [3]British Antarctic Survey, Cambridge, UK. [4]Institute of Marine Research, Tromsø, Norway. [5]National Institute of Oceanography and Applied Geophysics—OGS, Trieste, Italy. [6]University of Gothenburg, Gothenburg, Sweden. [7]Present address: School of Environmental Sciences, University of East Anglia, Norwich, UK. ✉e-mail: e.droste@uea.ac.uk

In more indirect ways, sea ice affects winter surface ocean properties through its dampening effect on wind-driven mixing with CDW[15,16]. Years with longer sea ice duration are associated with higher stratification and shallower mixed layer depths (MLDs) in winter (i.e. less vertical mixing) relative to years with shorter sea ice duration (i.e. more vertical mixing with CDW). CDW is a source of heat to the surface, and therefore the impact of sea ice on wintertime stratification has been shown to be a major driver in interannual variability of ocean heat flux[15,16]. CDW is also a source of dissolved inorganic carbon (DIC). To investigate whether sea ice has similar implications for interannual variability in wintertime surface $fCO_2$ compared to heat flux, we need year-round marine carbonate chemistry observations.

Here, we draw on measurements from the Rothera Time Series (RaTS) in Ryder Bay, on the West Antarctic Peninsula[17] (Fig. 1). Along with a few others (e.g., refs. 18,19), this is a rare dataset with in situ, year-round observations of key oceanographic and biogeochemical variables[20]. The time series commenced in 1997 and consists of conductivity-temperature-depth (CTD) profiles with concurrent discrete seawater samples at 15 m depth for the analysis of diverse properties, including nutrients since 1998[17], seawater oxygen isotopes since 2002[21], DIC, and total alkalinity (TA) since 2010[3]. The latter variables allow us to compute seawater $fCO_2$, from which we can determine the difference with the atmosphere (i.e. $\Delta fCO_2$) and sea-air $CO_2$ flux between the end of 2010 and the beginning of 2020. We draw on this rare year-round, interdisciplinary dataset to derive a process understanding of interannual variability in the winter marine carbonate system, its drivers, and the role that sea ice cover plays in its regulation.

## Interannual variability of wintertime $\Delta fCO_2$ and sea-air $CO_2$ flux

The mean annual $CO_2$ flux (from start of summer to end of spring) is negative for every year in the RaTS, ranging between −0.9 and −0.1 mol m$^{-2}$ year$^{-1}$. This means that Ryder Bay is a small net annual $CO_2$ sink[3]. It is the result of strong $CO_2$ uptake in summer (15 December–5 March) and autumn (negative $CO_2$ flux) overcompensating for the outgassing, which mostly occurs in winter (1 June–8 October) and spring

(positive $CO_2$ flux; Fig. 2a). The strong seasonal cycle is also reflected in the $\Delta fCO_2$ (Fig. 2b). This seasonality is tightly coupled to changes in seawater temperature and salinity (Fig. 2c, d). Solar radiation and meltwater input (low in DIC) from sea ice and surrounding glaciers create a highly stratified, fresh, and relatively warm surface layer in summer. This decreases $fCO_2$ through dilution, although the dominant driver of annual summer $fCO_2$ decrease is biological productivity[11].

As winter approaches, the temperature of surface seawater decreases and summer stratification is eroded by deepening of the mixed layer[16,22]. In Ryder Bay, seasonal vertical mixing is mostly attributed to mechanical mixing by surface wind stress, rather than convection by brine rejection from local fast sea ice formation[16,22]. The deepening mixed layer entrains modified-CDW (mCDW), a form of CDW from the southern boundary of the Antarctic Circumpolar Current that accessed the shelf primarily via deep, glacially-scoured canyons[23], and cross-shelf transport facilitated by eddies (Fig. 1a)[24,25]. mCDW is rich in DIC, which during vertical mixing increases $fCO_2$ at the surface to levels exceeding atmospheric $fCO_2$ in most years. This changes the sign of $\Delta fCO_2$ from negative to positive, and could lead to release of $CO_2$ to the atmosphere. However, sea-air $CO_2$ flux is strongly attenuated by sea ice. $CO_2$ outgassing in winter in Ryder Bay is therefore largely precluded by its sea ice cover. To account for this effect of sea ice on $CO_2$ flux, we linearly scale our flux calculations with the fraction of open water.

Superimposed on its strong seasonal carbon cycle, Ryder Bay shows substantial interannual variability in $CO_2$ flux, particularly in the amount of outgassing occurring each winter (Fig. 2a). This is not only the result of year-to-year differences in sea ice cover and its effect on gas exchange, but also of interannual variability in $\Delta fCO_2$ (Fig. 2b).

The mean interannual variability of $\Delta fCO_2$ is mainly explained by its variability in winter, as suggested by the significant relationship between winter and annual means of $\Delta fCO_2$, with an $R^2$ of 0.86 ($p < 0.05$; Fig. 3b). Mean spring $\Delta fCO_2$ also has a significant relationship to annual means ($R^2 = 0.82$; Fig. 3c), because the seawater $fCO_2$ levels at the end of winter determine the starting point from which springtime processes, such as primary productivity and meltwater dilution, reduce seawater $fCO_2$. While

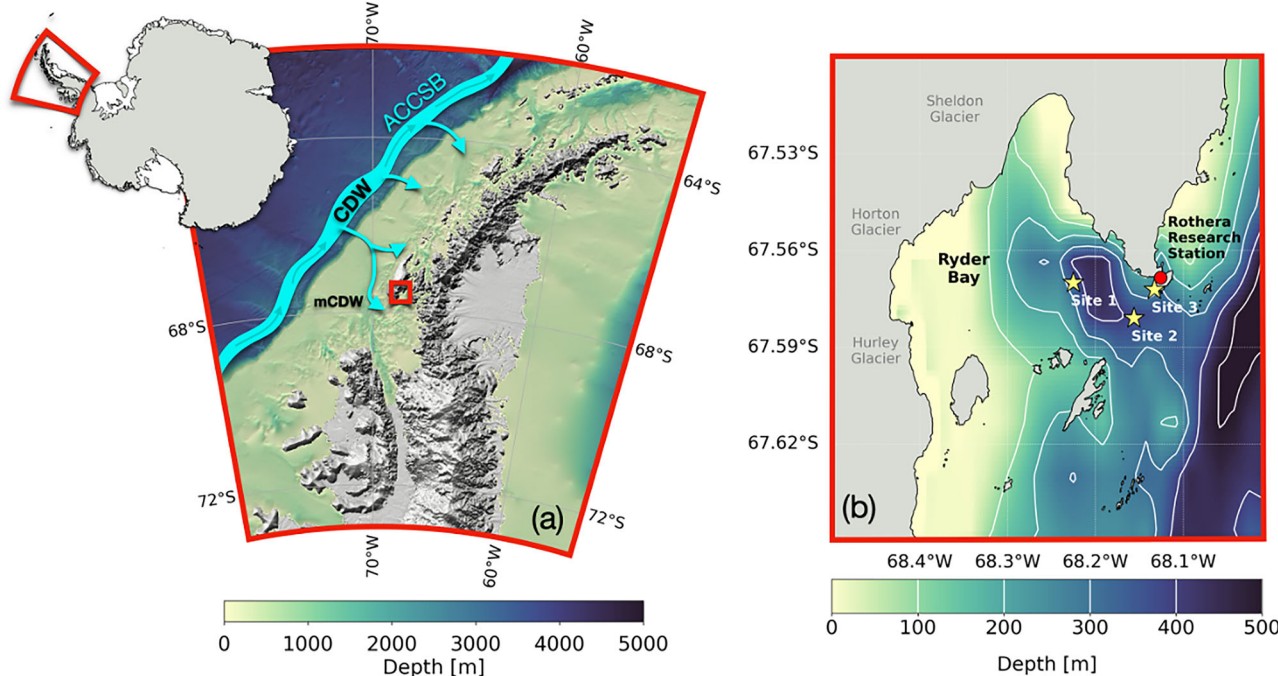

**Fig. 1 | Map of the study region and sampling sites in Ryder Bay. a** West Antarctic Peninsula region with bathymetry[72]. Blue band indicates the Antarctic Circumpolar Current Southern Boundary (ACCSB) transporting Circumpolar Deep Water (CDW) northwards[75]. Smaller arrows indicate CDW accessing the shelf, where it

becomes mCDW (modified-Circumpolar Deep Water). The red square indicates Ryder Bay, which is shown in greater detail in **b**. In **b**, also the RaTS sampling sites 1 (67°34.20′S, 68°13.50′W), 2 (67°34.85′S, 68°9.34′W), and 3 (67°34.33′S, 68°7.97′W) are shown.

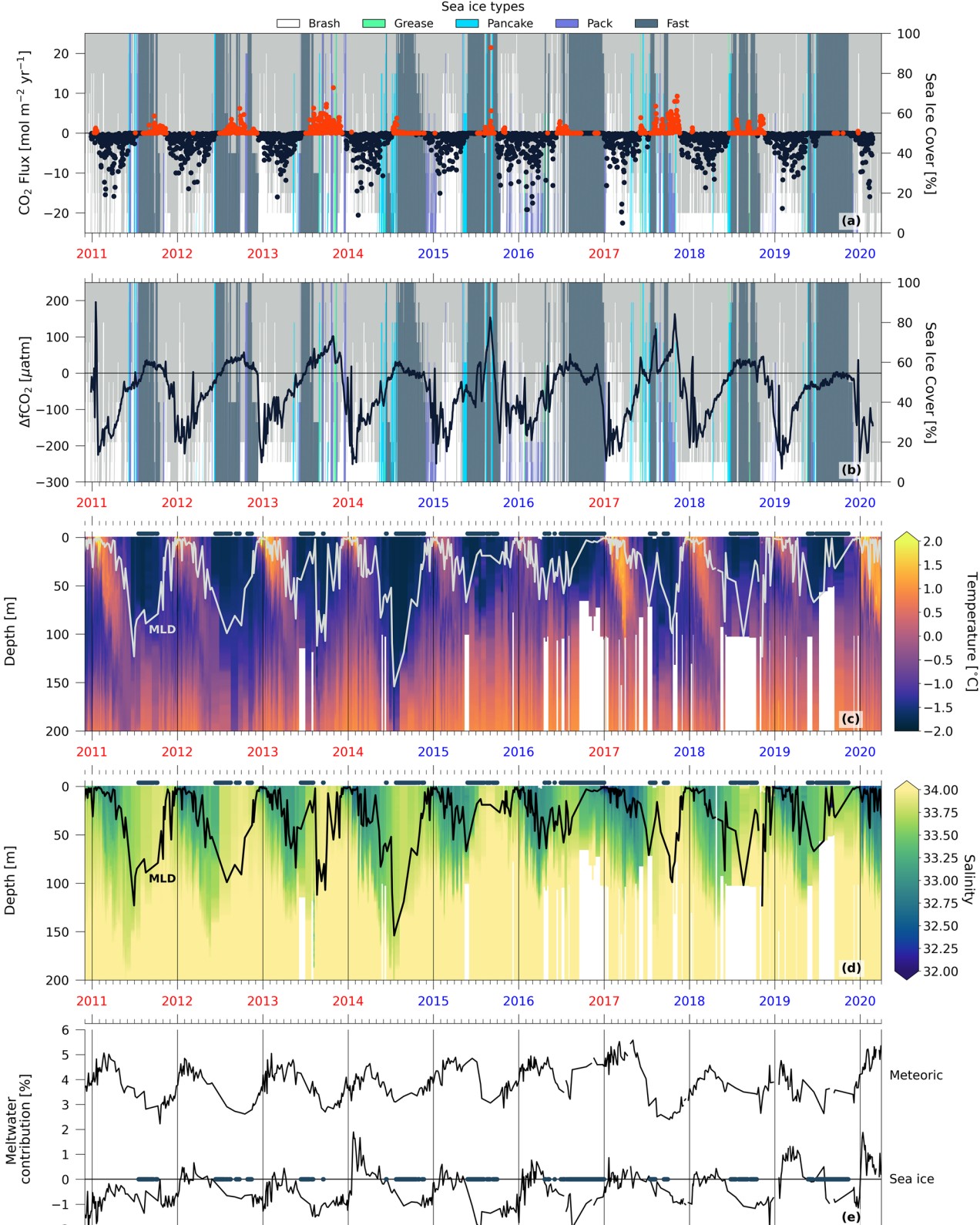

**Fig. 2 | The Rothera Time Series of CO₂ flux, ΔfCO₂, and physical variables in Ryder Bay between 28 December 2010 and 28 February 2020. a** Sea-air CO₂ flux in Ryder Bay. Positive (red) and negative (black) fluxes indicate sea-air CO₂ release to the atmosphere and uptake, respectively. Sea ice cover and type are shown in the background. High- or low-stratification years are marked in blue and red, respectively. **b** ΔfCO₂ in Ryder Bay. **c** Temperature in the upper 200 m of Ryder Bay. Mixed layer depth (MLD) is indicated by a white line. **d** Same as (**c**), but for salinity and MLD in black. **e** Meltwater contribution of meteoric and sea ice origin. Markers in **c–e** indicate days when fast sea ice cover was ≥80%.

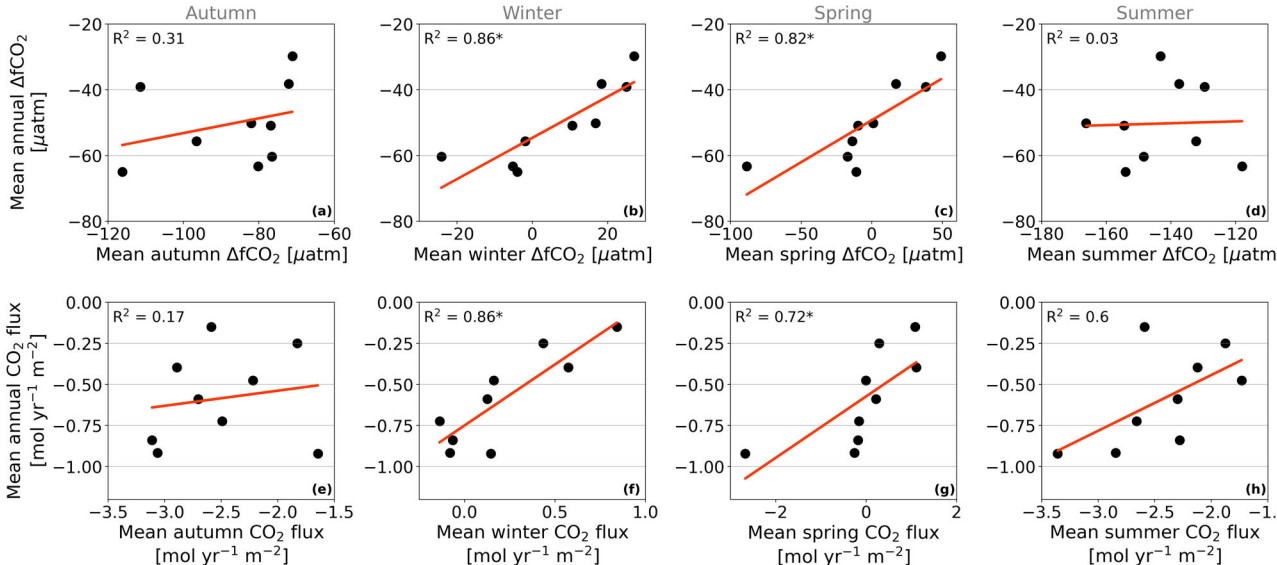

**Fig. 3 | Relationships between annual and seasonal means for ΔfCO₂ and fCO₂ flux in Ryder Bay. a–d** Relationship between mean annual ΔfCO₂ and the mean of each season: autumn, winter, spring, and summer, respectively. The annual mean is based on data between the start of summer on 15 December and the end of spring on 14 December the following year. The linear regression line is shown in red. Significance ($p < 0.05$) on the relationship is indicated by an asterisk next to the $R^2$ value. **e–h** Same as (**a–d**), but for CO₂ flux.

the strongest sea-air CO₂ fluxes occur in summer and autumn (i.e. when sea ice cover is low), their mean ΔfCO₂ shows no significant relationship to their annual means ($p > 0.05$; Fig. 3a, d). We derive a similar conclusion for the CO₂ flux (Fig. 3e–h). Thus, winter explains most of the interannual variability of CO₂ flux, despite substantial year-to-year variability in mean summertime values and the important role of summertime processes in net atmospheric CO₂ uptake. On a larger scale, this is consistent with previous findings that deep water mass ventilation in winter governs interannual variability of the Southern Ocean CO₂ sink[26]. Understanding wintertime processes is therefore key to comprehending the overall interannual variability of not only the marine carbonate system, but also of sea-air CO₂ exchange in this region.

**Impact of sea ice cover on stratification and ΔfCO₂**

To explain the varying mean ΔfCO₂ in winter, we turn to the dominant source of DIC. mCDW supplies DIC to the surface through vertical mixing. The extent of winter vertical mixing in Ryder Bay has been shown to be strongly associated with sea ice cover, which dampens wind-driven mixing and thus supports relatively high-stratification in winter[15,16]. Thus, years with winter fast ice that lasts longer tend to have a higher mean winter stratification and shallower mixing (MLD < 50 m) than years with shorter-lasting winter fast ice, which are characterised by deeper mixing and MLD > 50 m[16].

To investigate the effect of wintertime stratification on carbonate chemistry near the surface, we have characterised winters as high- or low-stratification, depending on whether mean winter stratification in the top 100 m is above or below 1500 J m⁻², respectively[16]. According to this threshold, the winters that are considered to have relatively high-stratification are those of 2015, 2016, 2018, and 2019 (number of seawater samples, $n = 212$), while those that are considered to have relatively low-stratification are the winters of 2011, 2012, 2013, 2014, and 2017 ($n = 283$). This categorisation has also been applied to all other years since 1998, for which CTD data are available; 2000, 2001, 2005 and 2020 are excluded here due to lack of measurements at the end of winter and/or during spring[16]. We subsequently compare the mean seasonal progression for physical and marine carbonate system variables between high- and low-stratification years.

Consistent with previous work on Ryder Bay[16], years with a short duration of fast ice (i.e. number of days with at least 80% fast ice in autumn

and winter combined; Fig. 4b) are characterised by low-stratification in winter (Fig. 4c) leading to deep mixed layers (Fig. 4d). Salinity increases at 15 m depth (Fig. 4e) as a result of mixing with mCDW (Fig. 4f). The latter is quantified using a three-component mass-balance method with regional end-members of salinity and seawater oxygen isotopes for mCDW, sea ice meltwater, and meteoric water[11,21].

In contrast, years with longer-lasting winter fast ice are characterised by higher stratification in winter and shallower mixed layers (<50 m). In these high-stratification conditions, mCDW contribution and associated salinity reach lower maximum values and start decreasing mid-winter, instead of at the end of winter as seen in the low-stratification years (Fig. 4e, f). This earlier seasonal decrease in salinity is attributed to the onset of increased meteoric water input, the source of which is meltwater from glaciers surrounding Ryder Bay, plus any direct precipitation. The signal of meteoric water increases mid-winter in high-stratification years, but only at the end of winter in low-stratification years (Fig. 4g). While the volume of meteoric water discharged to the ocean is not itself dependent on ocean stratification, its prevalence in the near-surface layers will be affected by the depth of the mixed layer over which the meltwater is vertically spread. A shallower mixed layer will typically reflect higher meteoric water contributions and will support faster re-stratification at the end of winter.

Although more nuanced, the surface marine carbonate chemistry responds to winter stratification similarly to salinity and this leads to important differences in the mean ΔfCO₂ between high- and low-stratification years. In high-stratification years, DIC and TA start decreasing earlier in the winter compared to low-stratification years (Fig. 5a, b). This is because meteoric water dilutes DIC and TA in the relatively shallow mixed layer. As a result, in high-stratification years, ΔfCO₂ decreases sharply before the end of winter, such that its mean becomes negative (i.e. potential atmospheric CO₂ uptake; Fig. 5c) before spring starts. In contrast, in low-stratification years, ΔfCO₂ remains positive (i.e. potential CO₂ outgassing) well into the spring.

The sign of ΔfCO₂ is particularly relevant to sea-air CO₂ exchange in this part of year, i.e. in the transition from winter to spring, because it coincides with the time when sea ice retreats. In low-stratification years, fast sea ice has usually already retreated before the start of spring, allowing seawater to release CO₂ to the atmosphere (Fig. 5d). Conversely, in most high-stratification years, sea ice cover is more persistent, lasting well into

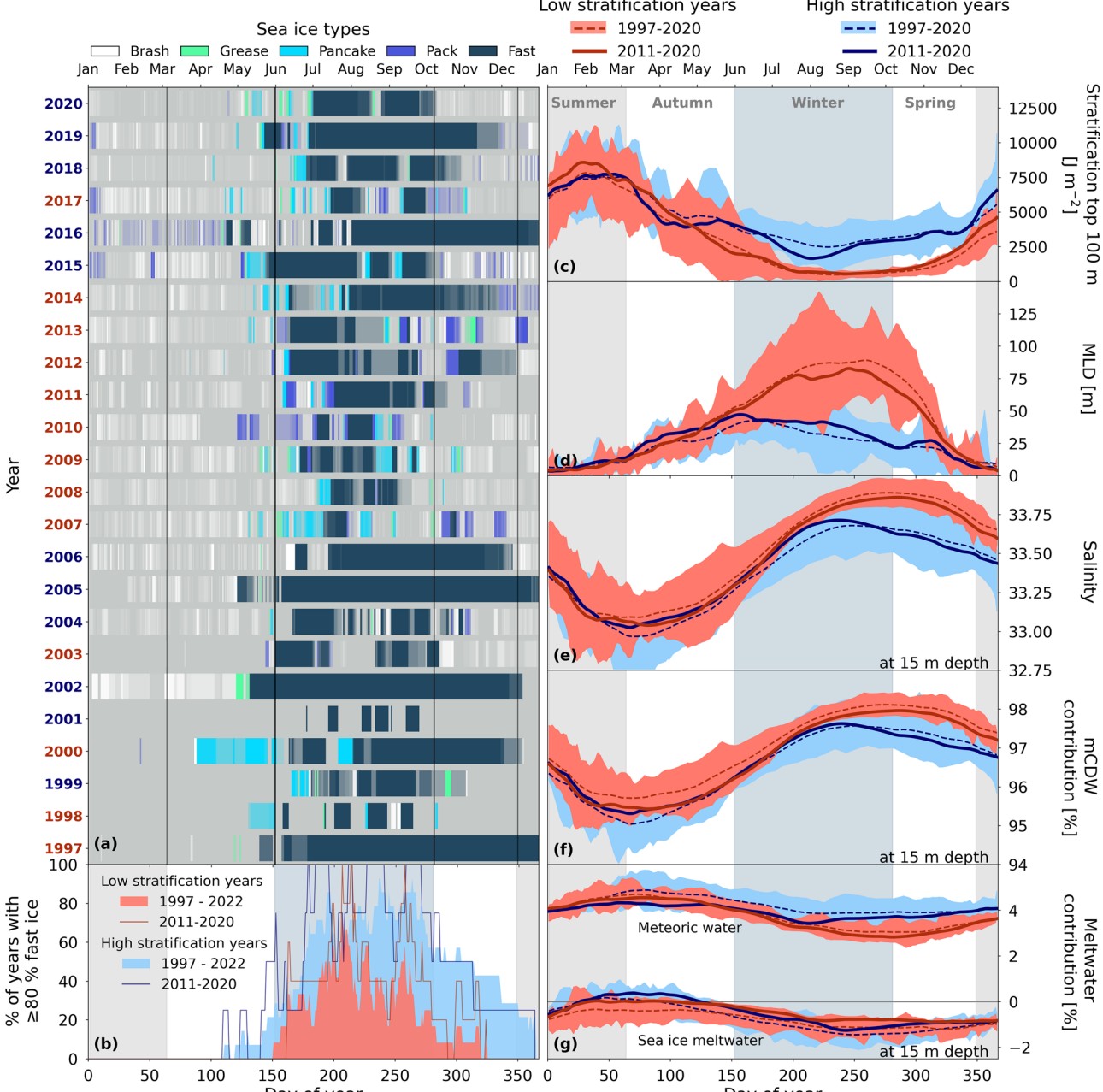

**Fig. 4 | Wintertime differences in ocean stratification between high- and low-sea ice years in Ryder Bay. a** Sea ice types in Ryder Bay between 1997 and 2020. Transparency gives a qualitative indication of the fraction of spatial sea ice coverage. High- or low-stratification years are marked in blue and red, respectively. Vertical lines separate the seasons. **b** Percentage of years with ≥80% fast sea ice cover on a day of year, calculated separately for high- or low-stratification years. We have included all available sea ice data up until 2022. **c–g** Mean stratification in the top 100 m, MLD, salinity, mCDW contribution, and meltwater contribution for high- and low-stratification years, respectively. Dashed lines indicate 30-day running means for 1997-2020, with filled areas indicating $1-\sigma$. Full lines are 30-day running means between 2011 and 2020. Filled vertical areas in the background indicate the seasons.

spring. When it eventually retreats, seawater $fCO_2$ is already low enough for the ocean to absorb atmospheric $CO_2$.

Sea ice, therefore, has a dual effect on sea-air $CO_2$ flux: it directly controls sea-air gas exchange, and indirectly regulates surface $\Delta fCO_2$. As a result, in high-stratification, high sea ice years Ryder Bay is a stronger annual atmospheric $CO_2$ sink by on average 20% (0.008 mol m$^{-2}$) compared to low-stratification, low-sea ice years (Fig. 5e). If we scale $CO_2$ flux by only fast sea ice instead of all sea ice types, high-stratification years annually absorb on average 27% (0.013 mol m$^{-2}$) more $CO_2$ than low-stratification years.

We are aware that biogeochemical processes, such as remineralisation of organic matter and calcium carbonate dissolution, could have affected

winter DIC and TA content and their variability. However, this contribution is likely to be small in winter relative to contributions of mixing[11]. While our findings highlight the relevance of physical wintertime processes on inter-annual variability of $CO_2$ uptake, we want to remark that other studies showed effects of wintertime stratification on both physical[16,27] and biogeochemical processes[17,28–30] in the following spring and summer. The net effect of these processes on marine carbonate chemistry during the productive season is likely complex. Whilst we are unable to explore these effects here, they are notable topics for future research.

We consider our estimates of net $CO_2$ uptake for Ryder Bay to be conservative, given that measurements at 15 m depth might not be

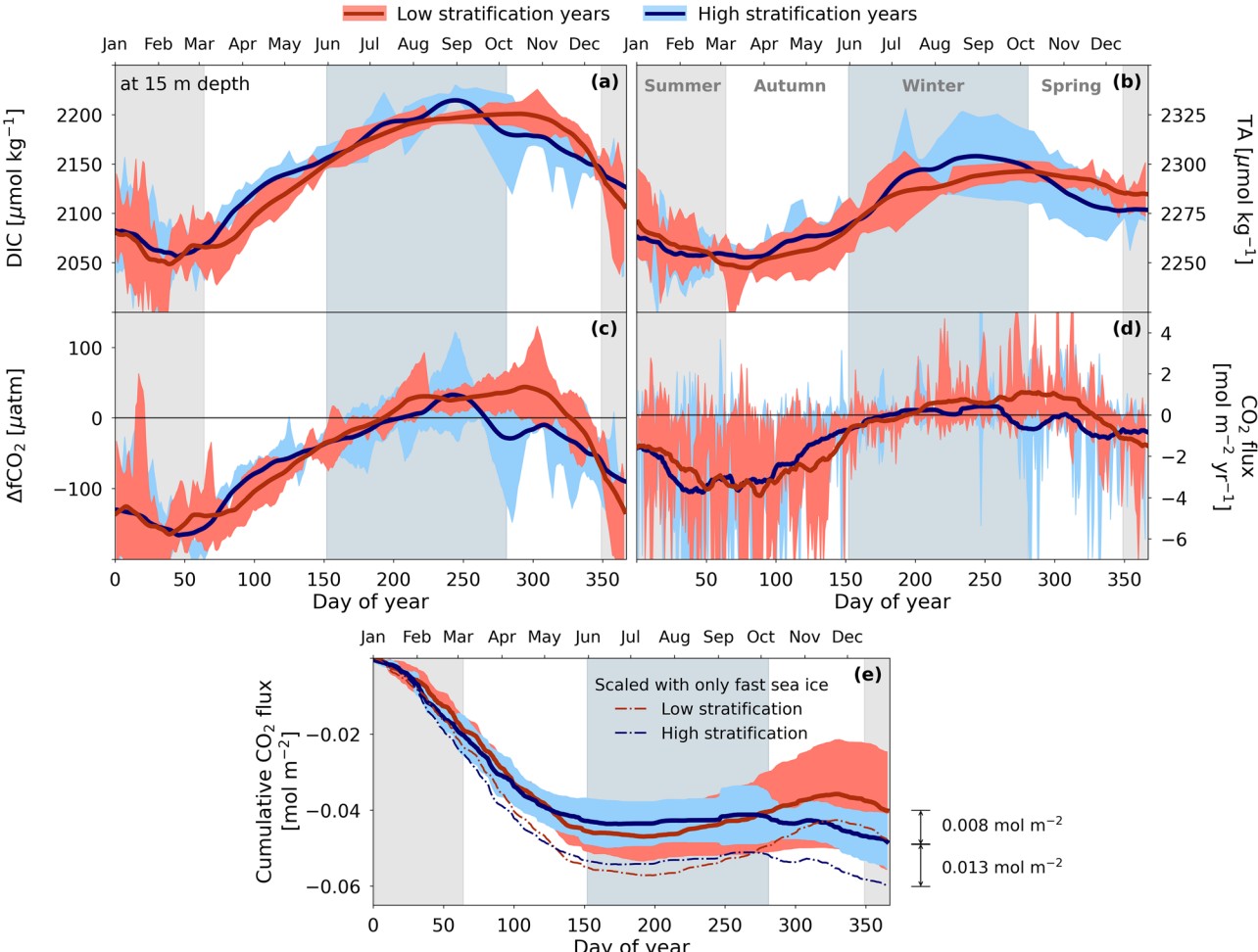

**Fig. 5 | Wintertime differences in surface marine carbonate system between high- and low-stratification years in Ryder Bay.** Mean DIC (**a**), TA (**b**), $\Delta fCO_2$ (**c**), $CO_2$ flux (**d**) and cumulative $CO_2$ flux (**e**) for low- and high-stratification years between 2011 and 2019. Filled areas indicate $1-\sigma$. Thick lines are 30-day running means. Vertically filled areas in the background indicate the seasons.

representative of the surface properties throughout the year[3]. Strong vertical gradients in the upper 15 m may occur in the melting season due to biological activity and meltwater input, resulting in overestimation in surface fCO_2 and underestimation in summer atmospheric $CO_2$ uptake[3,31,32]. We expect this limitation to be applicable to the summer months of all years, regardless of whether the winters are classified as high- or low-stratification. This limitation on the $CO_2$ flux estimates is therefore expected to have minimal effect on the relative differences in $CO_2$ uptake between high- and low-stratification winters.

Given the important role of sea ice on the interannual variability of sea-air $CO_2$ flux, we would here like to highlight a link to modes of large-scale atmospheric variability, such as the Southern Annular Mode (SAM) and the El Niño Southern Oscillation phenomenon. The former is broadly characterised by an oscillation in surface atmospheric pressure between a node over Antarctica and a ring over lower latitudes, with consequent fluctuations in the dominant westerly winds overlying the Southern Ocean, whilst the latter is focused on variability emanating from the equatorial Pacific Ocean, but with high-latitude impacts[33,34]. A positive SAM occurred in 2015, followed by an El Niño event. The pairing of these two phenomena caused anomalous atmospheric cooling in winter along the West Antarctic Peninsula[35], resulting in prolonged sea ice cover in both 2015 and 2016. According to the mechanisms discussed in the current study, this explains higher wintertime stratification, lower mean winter $\Delta fCO_2$, and higher annual atmospheric $CO_2$ uptake in Ryder Bay compared to years with less sea ice cover. Modes of atmospheric variability clearly play an important role

in the interannual variability of sea-air $CO_2$ flux along the West Antarctic Peninsula. Their key role for open ocean carbon uptake has further been demonstrated by other studies (e.g., refs. 36–39), indicating that wind intensity associated with climate modes regulates wintertime deep water ventilation and therefore the interannual variability of the Southern Ocean $CO_2$ sink[26]. Given that the conditions in Ryder Bay are inherently local, they cannot be taken as indicative of the broader West Antarctic Peninsula or Southern Ocean[16,40]. However, by using these locally-collected, year-round observations, we have been able to explore mechanisms behind ocean $CO_2$ uptake variability, including the role of sea ice and wintertime processes, which remain largely unexplored for other parts of the Southern Ocean, especially within seasonally sea ice-covered regions.

## Conclusion
The apparent regime shift of Antarctic sea ice cover since 2016[41,42] has increased the urgency to better understand wintertime $\Delta fCO_2$ variability and its association with sea ice. Based on year-round measurements of the RaTS, we have determined that sea ice cover plays a crucial role in the interannual variability of mean $\Delta fCO_2$ and sea-air $CO_2$ flux by mediating winter stratification.

In regions where wind stress, rather than brine-driven convection, is the dominant driver of vertical mixing, the presence of sea ice can have a dampening effect[16]. Therefore, years with relatively delayed or little sea ice cover are associated with winters with lower stratification and deeper mixed layers compared to high sea ice years (Fig. 6a). Over the winter season,

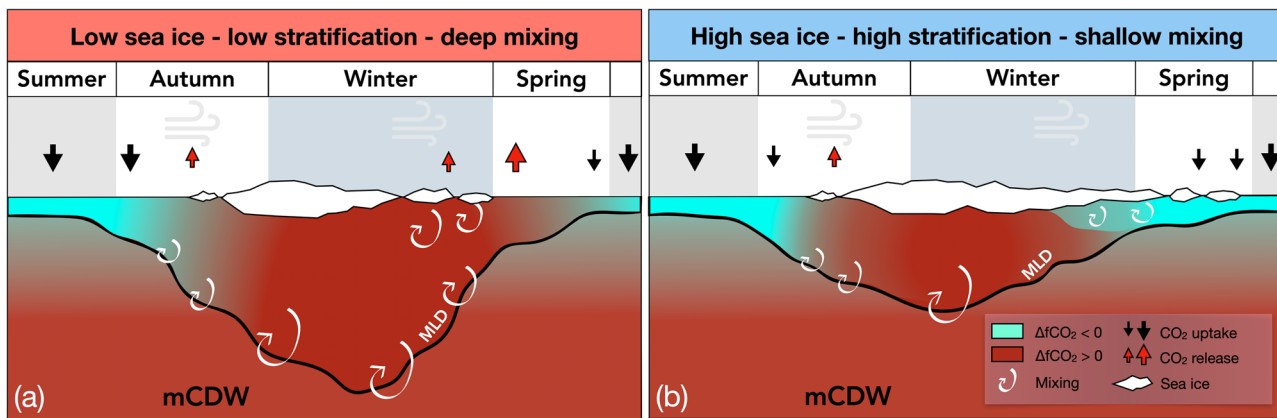

**Fig. 6 | Schematic representation of the effect of sea ice cover on wintertime ocean stratification, ΔfCO₂, and CO₂ flux. a** Low-sea ice conditions lead to deep wind-driven mixing, low-stratification in winter, and high ΔfCO₂, resulting in CO₂ outgassing to the atmosphere at the end of winter and in spring. **b** Higher sea ice conditions lead to higher wintertime stratification, and lower ΔfCO₂, resulting in less CO₂ outgassing and higher annual net atmospheric CO₂ uptake by the ocean, compared to years with low-stratification winters.

deeper mixed layers entrain more DIC from carbon-rich subsurface waters. This process maintains a high and positive ΔfCO₂, leading to outgassing as soon as sea ice retreats. Positive feedback processes can lead to enhanced CO₂ outgassing in subsequent years, because ocean heat uptake is typically enhanced in summers following winters with short sea ice duration[16], thereby delaying sea ice advance[15]. Conversely, years with long sea ice duration have a shallower winter mixed layer that re-stratifies earlier in the year (Fig. 6b). Dilution by meteoric water is stronger in shallower mixed layers, which drives surface ΔfCO₂ down to negative values before sea ice retreats and minimises CO₂ outgassing.

Previous studies have explored how sea ice shapes the seasonal variability of sea-air CO₂ flux by dampening wintertime CO₂ outgassing and enhancing summertime CO₂ uptake, particularly for coastal regions[11,43]. Substantial interannual variability has also been observed in previous work[43–45], but how this variability is affected by sea ice is not yet fully understood. To develop a better understanding, our unique case study on Ryder Bay suggests that ice-ocean feedbacks on ocean-interior processes need to be considered, especially in winter. These findings complement similar conclusions for interannual variability in ocean stratification[16], phytoplankton blooms and biomass[17,28], phytoplankton distribution in the water column[27], community composition[28,29] and nutrient cycling[30].

The coastal regions of Antarctica, such as Ryder Bay, collectively constitute an important contribution to the overall Southern Ocean CO₂ flux[3,19,43,46]. Hence, we need to understand the key mechanisms and feedback processes operating here. Whilst open ocean conditions across the broad Southern Ocean are necessarily different from coastal systems, identification of important mechanisms in such coastal regions can inform future research priorities more widely. In particular, our results reveal the importance of wintertime processes in explaining interannual variability in the marine carbonate system, which is supported by open Southern Ocean studies that suggest strong links between wintertime stratification and carbon cycling[47]. Assessments of net CO₂ exchange based predominantly on summer data are therefore unlikely to resolve critical processes affecting the interannual variability of Southern Ocean CO₂ uptake. Our findings emphasise a need to expand and sustain year-round observations of dynamic ice-ocean interactions in seasonally sea ice-covered regions to better capture their role in a changing polar environment.

## Methods
### Measurements and sampling
Site 1 in Ryder Bay (67°34.20′S, 68°13.50′W) is the primary sampling location for the RaTS, where the maximum water depth is about 500 m (Fig. 1b). Sites 2 (67°34.85′S, 68°9.34′W) and 3 (67°34.33′S, 68°7.97′W) are used as alternatives, when sea ice cover and/or weather conditions are unfavourable. All sampling sites are representative of the processes regulating the surface layer carbonate system of Ryder Bay[11]. CTD casts, which started in 1998[17], are deployed approximately weekly from the side of a small boat in the melting season and every 1–2 weeks during the winter season, or whenever weather and sea ice conditions permit. Casts were deployed through a man-made hole in the ice when land-fast ice covered the sampling sites. Details on sampling at Rothera and methods have been published in previous work[20].

MLD is calculated as the depth at which the density is greater than that at a surface reference depth (10 m depth) by 0.05 kg m$^{-3}$ [11,48,49]. Stratification is defined in a bulk sense as the additional energy that would be required if the water in a certain depth range were homogenised by mixing[16,50].

Daily sea ice cover and type (i.e. brash, grease, pancake, pack, or fast ice) were visually recorded daily by personnel at the Rothera Research Station[20]. Fast sea ice days are defined as days when fast sea ice cover is at 80% or higher, and they are used in figures to illustrate high- and low-sea ice years in Ryder Bay.

CTD casts profiled the top 200 m of the water column until 2003, after which full-depth casts were deployed, i.e. down to 500 m for Site 1, to 300 m for Site 2, and to 100 m for Site 3[20]. Seawater sampling was done directly after the deployment of the CTD cast, using a Niskin bottle. The standard sampling depth of seawater is 15 m. Dissolved inorganic nutrients (nitrate plus nitrite, phosphate, silicate) have been collected since the start of the RaTS in 1998[17,20]. The time series for stable seawater oxygen isotope samples ($\delta^{18}O$; the standardised ratio of stable oxygen isotopes in seawater, $H_2^{18}O$ to $H_2^{16}O$) started in 2002[20,21,48].

Sample collection for DIC and TA started on 28 December, 2010[3]. While the time series is ongoing, the final data point for DIC/TA included in this study was collected on 28 February 2020. DIC and TA samples were collected in 250 mL borosilicate glass bottles, fixed with mercuric chloride at Rothera Research Station, and sealed onsite for later analysis, according to Standard Operating Procedure (SOP) 1 in Dickson et al.[51]. Each 250 mL sample allowed for one DIC and one TA measurement; i.e. no repeat measurements could be done on the same sample. Occasionally, two samples were collected at the same depth from the same cast, from which an average would be calculated for the final results. The total number of DIC and TA data points at 15 m depth used in this study is 585 and 570, respectively, with an average of about 64 samples per year for DIC and 62 for TA. Most samples were analysed at the University of East Anglia, UK, using a VINDTA 3C (Versatile INstrument for the Determination of Total inorganic carbon and Alkalinity) system[52], which includes a coulometer for the measurement of DIC following SOP 2 in Dickson et al.[51], and a Titrino® for the potentiometric titration for TA following SOP 3b. TA was determined using the Calkulate Python package version 23.2.2[53]. Samples

collected from the end of 2013–2016 were analysed using a VINDTA 3C onsite at Rothera Research Station, following the same SOPs.

Over the years, various batches of the Certified Reference Material (CRM) from A. Dickson's laboratory at the Scripps Institution of Oceanography (U.S.A.) were used to calibrate the VINDTA instruments and to derive the analytical uncertainties represented by 1 $\sigma$ per CRM batch, which are <2.7 µmol kg$^{-1}$ for DIC and TA with the exception of data between 2016 and 2017, which had higher uncertainties of <5.4 µmol kg$^{-1}$.

### fCO₂ and CO₂ flux calculations

We calculated fCO₂ using the PyCO2SYS Python package version 1.8[54], based on Lewis et al.[55], with data inputs for DIC, TA, phosphate, silicate, temperature, salinity, and pressure. We use the dissociation constants for bisulphate from Dickson et al.[56], for hydrogen fluoride from Dickson et al.[57], the boron-salinity relationship from Uppström et al.[58], as recommended by best practice[59]. We used the carbonic acid dissociation constants by Goyet & Poisson et al.[60] to be consistent with previous work[3,11]. The mean propagated uncertainty for fCO₂, from the analytical uncertainties of DIC and TA, is 8 µatm, as calculated by PyCO2SYS[54].

The sea-air CO₂ flux ($F_{CO_2}$) is calculated according to Eq. (1).

$$F_{CO_2} = \Delta fCO_2 \cdot K_0 \cdot (k \cdot (1 - f_{sic})) \tag{1}$$

$$\Delta fCO_2 = fCO_{2_{sw}} - fCO_{2_{atm}} \tag{2}$$

$\Delta fCO_2$ is the difference between the fCO₂ in the seawater ($fCO_{2_{sw}}$) and the atmosphere ($fCO_{2_{atm}}$) at the sea-air interface (Eq. (2)). Atmospheric $fCO_{2_{atm}}$ (in µatm) is determined from the atmospheric mole fraction of CO₂ in dry air (xCO₂) measured in air samples collected at Palmer Research Station on Anvers Island[61]. These were smoothed using a weighted linear least-squares regression method (Lowess, using Statsmodels Python Package) that weighed 30% of the data around each data point, following previous work on the RaTS[3]. xCO₂ is converted to fCO₂ using the method by Weiss et al.[62], including the second virial coefficient given in Weiss et al.[63] and atmospheric pressure data measured at Rothera[64]. This method requires a conversion of the dry mole fraction to the mole fraction in water vapour-saturated air, for which we used the water vapour pressure calculation by Weiss et al.[62]

$K_0$ is the CO₂ solubility and is a function of temperature and salinity, and is determined according to the parameterisation in Weiss et al.[63] that calculates $K_0$ in units of mol kg$^{-1}$ atm$^{-1}$. Multiplication with density converts the units to mol m$^{-3}$ atm$^{-1}$.

$k$ is the gas transfer velocity (in cm h$^{-1}$) and is calculated according to Wanninkhof et al.[12] and Ho et al.[65]. It depends on the Schmidt number, for which we use the parameterisation by Wanninkhof et al.[12], and has a quadratic relationship to wind speed, for which we use daily averages of continuous measurements at Rothera[64]. To match this daily resolution, seawater temperature, salinity, and seawater fCO₂ were linearly interpolated between sampling days. Wind speed is measured at 42 m above sea level. We use Eq. (3) to adjust the daily averaged wind speed to the reference height at 10 m above sea level, as is appropriate for the $k$ parameterisation. The formulation for the wind speed adjustment is taken from the COARE model[66]. The term $z_h$ is the height of the wind speed measurement, $z_{10}$ denotes the reference height at 10 m, $U_h$ is the measured wind speed at height $z_h$, and $U_{10}$ is the wind speed adjusted to the reference height of 10 m above sea level.

$$U_{10} = U_h \cdot \frac{\log_{10}\left(\frac{z_{10}}{10^{-4}}\right)}{\log_{10}\left(\frac{z_h}{10^{-4}}\right)} \tag{3}$$

Sea-air CO₂ flux is strongly attenuated by the presence of sea ice, which we account for by linearly scaling the gas transfer velocity, $k$, with the fraction of open water, determined in Eq. (1) by $(1 - f_{sic})$, where $f_{sic}$ represents the fraction of sea ice cover[3]. We use the sea ice cover of all ice types when scaling $k$, unless otherwise specified. Following a study on the first three

years of the Rothera carbonate chemistry time series[3], at 100% sea ice cover, we set $f_{sic}$ to 0.99 instead of 1. This means that $k$ is only ever scaled down by a maximum of 99%, allowing for a small amount of CO₂ flux to occur at all times[3,14,19,67]. This minimum flux represents the possibility for sea-air gas exchange through cracks in the ice, as well as sea ice-air gas exchange observed in previous studies[68–71]. However, the effect of this $k$ attenuation limit on total CO₂ flux is negligible (<0.2 mol m$^{-2}$ year$^{-1}$), likely because $\Delta fCO_2$ is relatively low when sea ice cover is 100%.

### Contributions of different water masses

Salinity and $\delta^{18}O$ are both conservative tracers and share a number of processes that affect them in similar ways, e.g. evaporation and precipitation. The great utility of $\delta^{18}O$ in polar waters is that it is only minimally affected by sea ice formation and melt, whereas salinity is very strongly affected by these processes. Conversely, high-latitude meteoric water input (glacial melt and net precipitation combined) strongly affects both $\delta^{18}O$ and salinity. This decoupling of the two tracers allows calculation of different freshwater sources. This is achieved quantitatively using a three-component mass-balance method, which has been used in previous RaTS studies[11,21,48]. It consists of three mass-balance equations: one for mass conservation (Eq. 4), one for salinity (Eq. 5), and one for $\delta^{18}O$ (Eq. 6). Each contains a component for the mCDW, sea ice meltwater, and meteoric water. Following previous work[11,21], we assume a system where the mCDW is the main oceanic water mass source, which is annually diluted in the surface layer by sea ice and glacial melt, and annually made more saline at the surface by sea ice production. Solving these equations results in fractions of each of these water mass types that contributed to the seawater sample.

$$f_{mCDW} + f_{sim} + f_{met} = 1 \tag{4}$$

$$S_{mCDW} \cdot f_{mCDW} + S_{sim} \cdot f_{sim} + S_{met} \cdot f_{met} = S \tag{5}$$

$$\delta^{18}O_{mCDW} \cdot f_{mCDW} + \delta^{18}O_{sim} \cdot f_{sim} + \delta^{18}O_{met} \cdot f_{met} = \delta^{18}O \tag{6}$$

In Eqs. (4–6), the variable $f$ denotes the fraction of either the local version of mCDW in Ryder Bay (mCDW), sea ice meltwater (sim), or meteoric water (met). In Ryder Bay, the latter will mostly consist of glacial ice meltwater. S and $\delta^{18}O$ are the salinity and oxygen isotope ratio measured in the sample, respectively. S and $\delta^{18}O$ with subscripts represent the end-member values for each of the water mass types. For consistency, we use the same end-members as previous work on the RaTS[11]: salinity end-members are 34.62, 7, and 0 for mCDW, sea ice meltwater, and meteoric water, respectively; and $\delta^{18}O$ end-members are 0.08 ‰, 2.1 ‰, and −17 ‰ for mCDW, sea ice meltwater, and meteoric water, respectively. The uncertainty in the end-members is estimated to be ±1%[21]. The meteoric end-member harbours most uncertainty, as it combines glacial meltwater, runoff, and net precipitation, all of which can have different end-member values[21].

### Seasonal definitions

Asymmetric seasonal changes in the water column in Ryder Bay highlight a need for a Ryder Bay-specific definition of the seasons that we can use to look at the interannual variability of seasonal processes, as well as inter-seasonal dependencies[11,16,48]. We define the range of the winter season between day-of-year 152 and 281 (i.e. 1 June–8 October). The summer season starts on day 349 (15 December) and ends on day 64 (5 March). The choice of these day-of-years is based on the seasonal variability of a number of key variables relevant to seasonal changes in seawater properties, using RaTS data available since 1997, as follows. The onset of winter coincides with the time when seawater between 0 and 15 m depth (where we have most of the chemical time series data) becomes well mixed (i.e. no detectable salinity gradient in the upper 15 m). The seawater temperature gradient is minimal at this point. The end of winter is determined by the average of time points when the salinity at 15 m depth, the contribution of mCDW at 15 m depth, and the MLD have reached their maxima, and the contribution of

meteoric water has reached its minimum. The summer starts when daily mean air temperatures at 2 m above sea level continuously exceed 0 °C. The summer ends when air temperatures continuously remain below 0 °C. The end of summer is also characterised by a minimum in salinity and mCDW fraction, maxima in sea ice melt fraction and glacial melt fraction, and maximum seawater temperature at 15 m depth.

## Reporting summary

Further information on research design is available in the Nature Portfolio Reporting Summary linked to this article.

## Data availability

Access details for Rothera Time Series data, including CTD measurements, nutrient concentrations and sea ice observations, can be found in Venables et al.[20]. Bathymetry data are available on Pangaea[72]. Wind speed and atmospheric pressure data are made available by the British Antarctic Survey[64]. Atmospheric $CO_2$ mixing ratios measured in air samples collected at Palmer Research Station on Anvers Island are published on the NOAA Global Monitoring Laboratory Data Repository[61]. Rothera Time Series DIC and TA data between 2010 and 2014 have previously been published on NOAA National Centers for Environmental Information[3,73]. The full Rothera DIC/TA dataset until 2020, including data between 2010 and 2014, is publicly available on the UK Polar Data Centre[74].

## Code availability

Python package Calkulate[53] was used to determine seawater total alkalinity, and PyCO2SYS[54] was used to determine seawater $fCO_2$.

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

## Acknowledgements

We would like to thank the Ocean Scientists at Rothera Research Station for carefully collecting the seawater samples over the years. Thanks to Marcos

Cobas-Garcia for co-analysing the DIC/TA samples collected in 2014–2016. Extended thanks to N. Clargo, M. Fenton, S. Ossebaar, J. Bown, P. Rozema and S. Henley for assistance in sample collection and analysis in 2013–2017. The Rothera Time Series has been funded by NERC through a sequence of National Capability awards, most recently NE/Y006178/1 (PRESCIENT). E.S.D. was supported by the Natural Environment Research Council (NERC) through the EnvEast Doctoral Training Partnership (grant no. NE/L002582/1), and partly by funding from the European Union's Horizon 2020 research and innovation programme under grant agreement no. 821001 and the NERC PICCOLO award (NE/PO21395/1). D.C.E.B. was partly supported by the NERC PICCOLO award (NE/PO21395/1). E.M.J. was supported by the research programme 866.13.006 (partly) financed by the Netherlands Polar Programme at the Dutch Research Council (Nederlandse Organisatie voor Wetenschappelijk Onderzoek). E.M.J. gratefully acknowledges the Royal NIOZ and University of Groningen for the opportunity to conduct fieldwork at the Dirck Gerritsz and Bonner laboratories at Rothera from 2013 to 2017. DIC/TA samples collected between the end of 2013 and 2016 were analysed onsite at Rothera, and samples from 2016 until January 2017 were analysed at UEA. The participation of M.P.M. was partly funded by the award NE/W004933/1 (BIOPOLE). We acknowledge support by the Open Access publication fund of Alfred-Wegener-Institut Helmholtz-Zentrum für Polar-und Meeresforschung.

## Author contributions

E.S.D. led the manuscript, wrote the first draft, analysed the DIC/TA samples for 2017–2020 of the time series, processed the data for 2016–2020, and performed computational analyses as shown in this study. D.C.E.B. designed the concept of this work and is responsible for the continuation and logistics of the Rothera carbonate chemistry time series. H.J.V. contributed physical oceanographic, nutrient, and sea ice cover data for Ryder Bay and is responsible for the continuation and logistics of the Rothera Time Series. E.M.J. analysed and processed DIC/TA samples for 2014–2016. M.P.M. contributed to the analysis of the physical data and oxygen isotope data. G.D.O. helped formulate and develop the understanding of this work. M.H. helped formulate and develop the understanding of this work. O.J.L. helped formulate and develop the understanding of this work. G.A.L. co-analysed the DIC/TA samples for 2014–2016 and 2019–2020. B.Y.Q. helped formulate and develop the understanding of this work. All authors contributed to the paper.

## Funding

## Competing interests

The authors declare no competing interests.
