## [Transparent Peer Review file · Communications Earth & Environment]

Sea ice controls net ocean uptake of carbon dioxide by regulating wintertime stratification

Corresponding Author: Dr Elise Droste

Version 0:

Decision Letter:

Dear Dr Droste,

Your manuscript titled "Sea ice controls net ocean CO₂ uptake by regulating wintertime stratification" has now been seen by 2 reviewers, and we include their comments at the end of this message. Both reviewers find that your work is of interest and provides novel information, but some points are raised, mainly related to improving clarity regarding methodological details and expanding the Discussion. We are interested in the possibility of publishing your study in *Communications Earth & Environment*, but would like to consider your responses to these concerns and assess a revised manuscript before we make a final decision on publication.

We therefore invite you to revise and resubmit your manuscript, along with a point-by-point response that takes into account the points raised. Please highlight all changes in the manuscript text file.

Please submit your point-by-point responses as a separate file, distinct from your cover letter where you can add responses to the Editors' comments that you do not want to be made available to the reviewers. Word files are preferred. We recommend that any figures, tables or graphs that are included in the response to reviewers are also included in the main article or Supplementary Information.

Please use the following link to submit your revised manuscript, point-by-point response to the referees' comments (which should be in a separate document to any cover letter), a tracked-changes version of the manuscript (as a PDF file) and the completed checklist:

Link Redacted

We hope to receive your revised paper within six weeks; please let us know if you aren't able to submit it within this time so that we can discuss how best to proceed. If we don't hear from you, and the revision process takes significantly longer, we may close your file. In this event, we will still be happy to reconsider your paper at a later date, as long as nothing similar has been accepted for publication at *Communications Earth & Environment* or published elsewhere in the meantime.

Please do not hesitate to contact us if you have any questions or would like to discuss these revisions further. We look forward to seeing the revised manuscript and thank you for the opportunity to review your work.

Best regards,

Nadine Schubert, PhD
Editorial Board Member
Communications Earth & Environment
orcid.org/0000-0001-7161-7882

Alice Drinkwater, PhD
Associate Editor
Communications Earth & Environment

EDITORIAL POLICIES AND FORMATTING

Editorial Policy: [Policy requirements](https://www.nature.com/documents/nr-editorial-policy-checklist.pdf) (Download the link to your computer as a PDF.)

- Behavioural and social science
- Ecological, evolutionary & environmental sciences
- Life sciences

<https://www.nature.com/documents/nr-reporting-summary.zip>

Furthermore, please align your manuscript with our format requirements, which are summarized on the following checklist: [Communications Earth & Environment formatting checklist](https://www.nature.com/documents/commsj-phys-style-formatting-checklist-article.pdf)

and also in our style and formatting guide [Communications Earth & Environment formatting guide](https://www.nature.com/documents/commsj-phys-style-formatting-guide-accept.pdf) .

*** DATA: Communications Earth & Environment endorses the principles of the Enabling FAIR data project (<http://www.copdess.org/enabling-fair-data-project/>). We ask authors to make the data that support their conclusions available in permanent, publically accessible data repositories. (Please contact the editor if you are unable to make your data available).

All Communications Earth & Environment manuscripts must include a section titled "Data Availability" at the end of the Methods section or main text (if no Methods). More information on this policy, is available at <http://www.nature.com/authors/policies/data/data-availability-statements-data-citations.pdf>.

If a community resource is unavailable, data can be submitted to generalist repositories such as [figshare](https://figshare.com/) or [Dryad Digital Repository](http://datadryad.org/). Please provide a unique identifier for the data (for example a DOI or a permanent URL) in the data availability statement, if possible. If the repository does not provide identifiers, we encourage authors to supply the search terms that will return the data. For data that have been obtained from publically available sources, please provide a URL and the specific data product name in the data availability statement. Data with a DOI should be further cited in the methods reference section.

REVIEWER COMMENTS:

Reviewer #1 (Remarks to the Author):

The article addresses the issue of the importance of the influence of sea ice on the calculations of CO₂ flux in the Southern

Ocean, through data obtained in the coastal region of the western Antarctic Peninsula. The topic addressed is of great interest today and brings significant advances to the scientific community. The article is very well written and presents excellent figures. The discussion and conclusions are based on the results presented. The methodology requires some clarification so that it can eventually be reproduced. I recommend accepting the article after minor corrections. Some minor comments are indicated below.

- The abstract needs to be better worked on, there is a need to include quantitative aspects related to the main results of the CO₂ flux.
- Lines 62-67: the sentences need to indicate a reference.
- Line 97: be consistent in the citation of the unit throughout the text.
- Lines 120-122: there is a need for further clarification, as indicated in the Methods section it is not clear how the CO₂ fluxes were calculated (see last comments).
- Line 132: not clear, please clarify.
- Figure 4: check the years in the legend for panel b, should 2022 be 2020?
- Lines 154-159: did you verify the dominant modes of variability (ENSO, SAM)? A mention on this is relevant for the study area, as the high stratification years appear to be years of ENSO+, for example.
- Lines 217-227: Great to see the caveats indicated here. Perhaps also report something related to the verified study area (Ryder Bay), since this is a very localized investigation to be representative of large coastal areas of the Southern Ocean.
- Lines 250-254: This paragraph needs to be better elaborated, mainly to highlight the contribution that previous studies (e.g. 12, 30) have already made to the topic of the impact of sea ice on CO₂ flux estimates. An additional reference may also help in this discussion: <https://doi.org/10.1029/2022GB007518>, regarding the marine biogeochemistry close to the study area.
- Lines 257-259: This sentence should be attenuated, indicating other processes that may not have been captured by the data evaluated in this study.
- Line 314: include the value of the propagated error to obtain the estimate of CO₂ fugacity.
- Lines 335 and 341: do not need to abbreviate "asl".
- Lines 342-451: it is not clear how this scaling was performed. I suggest including more details and information, since the indicated reference (3) also does not make it clear how this adjustment is included in equation 1.

Congratulations on the work! It was a very pleasant read.

Reviewer #2 (Remarks to the Author):

Summary

The manuscript by Droste et al. is well-written and provides in-depth discussions. The study presents 10 years of surface water carbonate chemistry data and CTD profiles at the Rothera Time Series on the West Antarctic Peninsula (WAP), including continuous winter observations - a rare dataset for coastal Antarctic. The main finding highlights the significant influence of sea ice condition and water stratification in winter/spring on interannual variability in air-sea CO₂ fluxes. Specifically, reduced sea ice coverage and lower stratification in winter lead to increased CO₂ outgassing during the winter to spring transition, and a 20% reduction in annual carbon drawdown. This is an important and timely research topic, and I believe it would be of great interest to the research community as well as broader audiences for Comm. Earth & Env. interested in climate change and carbon cycling.

While the study has explored the physical processes, biological processes (photosynthesis and respiration) are major drivers of annual carbon flux in this region and should be further discussed and incorporated in the proposed mechanism. For instance, could under-ice or ice-edge phytoplankton growth contribute to the spring CO₂ drawdown observed in high stratified years? (Hague and Vichi 2021, Biogeosciences, 18:25-38). In low stratified years, would deeper winter mixing (resulting in higher nutrient availability) and a longer open water period enhance biological carbon drawdown when integrated over the growing season?

Detailed:

Line 53-54: Could you clarify the limitations here? Are they due to calibration issue or related to parameters that can only be measured through ship-based observations?

Line 121: linear scaling.

Line 286: specify the approximate depth range for the CTD casts.

Line 374 – 37: While already cited Legge et al., it would be helpful to list the end member values (i.e., salinity and delta 18O) used in the calculation.

Communications Earth & Environment is committed to improving transparency in authorship. As part of our efforts in this

direction, we are now requesting that all authors identified as 'corresponding author' create and link their Open Researcher and Contributor Identifier (ORCID) with their account on the Manuscript Tracking System prior to acceptance. ORCID helps the scientific community achieve unambiguous attribution of all scholarly contributions. You can create and link your ORCID from the home page of the Manuscript Tracking System by clicking on 'Modify my Springer Nature account' and following the instructions in the link below. Please also inform all co-authors that they can add their ORCIDs to their accounts and that they must do so prior to acceptance.

Version 1:

Decision Letter:

Dear Dr Droste,

Your manuscript titled "Sea ice controls net ocean CO₂ uptake by regulating wintertime stratification" has now been assessed by the editorial team. We are delighted to say that we are happy, in principle, to publish a suitably revised version in Communications Earth & Environment.

We therefore invite you to revise your paper one last time to edit your manuscript to comply with our format requirements and to maximise the accessibility and therefore the impact of your work.

EDITORIAL REQUESTS:

****Please take care to match our formatting and policy requirements. We will check revised manuscript and return manuscripts that do not comply. Such requests will lead to delays. ****

SUBMISSION INFORMATION:

OPEN ACCESS:

Communications Earth & Environment is a fully open access journal. Articles are made freely accessible on publication. For further information about article processing charges, open access funding, and advice and support from Nature Research, please visit <https://www.nature.com/commsenv/open-access>

Link Redacted

Best regards,

Alice Drinkwater, PhD
Associate Editor
Communications Earth & Environment
Consulting Editor
Communications Sustainability

Author's Responses to Reviewers' Comments on COMMSNV-24-3668-T

Sea ice controls net ocean CO₂ uptake by regulating wintertime stratification

Nature Communications Earth & Environment
April 2025

Dear Reviewers,

Thank you for taking the time to give constructive feedback on our manuscript, titled *Sea ice controls net ocean CO₂ uptake by regulating wintertime stratification*. In this document, we address each of your comments. Reviewers' comments are **highlighted in blue**, and have been numbered by the authors for easy reference. All references made to line numbers refer to the line numbers in the originally submitted manuscript, unless otherwise specified. Please also find attached the revised manuscript, as well as a document highlighting all changes made.

Kind regards,

Elise S. Droste

Dorothee C. E. Bakker, Hugh J. Venables, Elizabeth M. Jones, Michael P. Meredith, Giorgio Dall'Olmo, Mario Hoppema, Oliver J. Legge, Gareth A. Lee, Bastien Queste

Reviewer #1 (Remarks to the Author):

Summary

The article addresses the issue of the importance of the influence of sea ice on the calculations of CO₂ flux in the Southern Ocean, through data obtained in the coastal region of the western Antarctic Peninsula. The topic addressed is of great interest today and brings significant advances to the scientific community. The article is very well written and presents excellent figures. The discussion and conclusions are based on the results presented. The **methodology requires some clarification so that it can eventually be reproduced**. I recommend accepting the article after minor corrections. Some minor comments are indicated below.

1.1. Methodology requires some clarification so that it can eventually be reproduced.

We are very grateful for this positive and supportive feedback. We accept the requirement to clarify the methodology, and have done this in the revision – please see our responses to comments #1.5 and #1.14 for specific details on this.

Detailed comments

1.2. Abstract needs to be better worked on, there is a need to include quantitative aspects related to the main results of the CO₂ flux.

Thanks for this comment. We have amended the abstract to include quantitative aspects as recommended.

Original text:

“Carbon dioxide (CO₂) exchange between the Southern Ocean and the atmosphere is strongly seasonal, characterised by atmospheric CO₂ uptake in summer, which is partly offset by CO₂ outgassing in winter. Sea ice conditions can shift this seasonal balance, inducing interannual variability in the net CO₂ uptake of the Southern Ocean.

Using a decade (2010-2020) of unique, year-round marine carbonate chemistry observations from the Rothera Time Series (West Antarctic Peninsula), we show that the interannual variability in fugacity of CO₂ (fCO₂) is closely dependent on wintertime processes. In winter, the duration of sea ice cover controls water column stratification, which in turn acts as a gateway between the carbon-rich ocean interior and the surface layer. This indicates that previous assessments of net CO₂ exchange based predominantly on summer data will not resolve critical processes, or adequately resolve the magnitude of sea-air CO₂ interannual variability.”

Revised text:

“Sea-air carbon dioxide (CO₂) exchange in the Southern Ocean is strongly seasonal, with atmospheric CO₂ uptake in summer, which is partly offset by CO₂ outgassing in winter. Sea ice conditions can shift this seasonal balance, inducing interannual variability in the Southern Ocean CO₂ sink.

A decade (2010-2020) of unique, year-round marine carbonate chemistry observations from the Rothera Time Series (West Antarctic Peninsula), reveal that interannual variability in seawater fugacity of CO₂ is dependent on wintertime processes. Sea ice duration controls ocean stratification, which acts as a gateway to the carbon-rich ocean interior. Consequently, years with high mean winter stratification (>1500 Jm⁻²) due to persistent sea ice cover absorb, on average, 20% more CO₂ than years with weaker stratification and less sea ice in winter.

Assessments of net CO₂ uptake from predominantly summertime data are therefore unlikely to resolve critical processes, or adequately resolve the magnitude of sea-air CO₂ flux interannual variability.”

1.3. Lines 62-67: the sentences need to indicate a reference.

Done.

Original text:

“The strength of sea-air gas flux is positively related to wind speed (Wanninkhof et al., 2014), but is significantly attenuated by sea ice cover, which limits direct sea-air gas exchange. This restrictive effect of sea ice prevents a large fraction of winter CO₂ outgassing in seasonally sea ice covered regions, but will similarly prevent atmospheric CO₂ uptake if seawater fCO₂ is low. Changes in the duration of sea ice cover are therefore able to shift the seasonal balance between summer CO₂ uptake and winter outgassing.”

Revised text:

“The strength of sea-air CO₂ flux is positively related to wind speed (Wanninkhof et al., 2014), but is significantly attenuated by sea ice cover, which limits direct sea-air gas exchange (Bakker et al., 2008). This restrictive effect of sea ice prevents a large fraction of winter CO₂ outgassing in seasonally sea ice covered regions (Legge et al., 2015). Similarly, it can prevent atmospheric CO₂ uptake if seawater fCO₂ is low, as has been observed in the Arctic (e.g., Bates et al., 2006). Sea ice is therefore key in maintaining a net atmospheric CO₂ sink, because changes in its duration are able to shift the seasonal balance between summer CO₂ uptake and winter outgassing.”

1.4. Line 97: be consistent in the citation of the unit throughout the text.

Done.

1.5. Lines 120-122: there is a need for further clarification, as indicated in the Methods section it is not clear how the CO₂ fluxes were calculated (see last comments).

We have revised the text to clarify how we scale the CO₂ flux according to sea ice cover (see revised text below). To add further clarifications, we have also revised the corresponding text in the Methods section, as shown in our response to comment #1.14.

Original text:

“However, winter CO₂ outgassing in Ryder Bay is significantly attenuated by partial or full sea ice cover. We account for this effect in our CO₂ flux calculations by linearly scaling the flux with the fraction of open water.”

Revised text:

“However, sea-air CO₂ flux is strongly attenuated by sea ice. CO₂ outgassing in winter in Ryder Bay is therefore largely precluded by its sea ice cover. To account for this effect of sea ice on CO₂ flux, we linearly scale our flux calculations with the fraction of open water.”

1.6. Line 132: not clear, please clarify.

We have revised the sentence.

Original text:

“Mean spring $\Delta f\text{CO}_2$ also has a significant relationship to annual means ($R^2 = 0.82$; Fig. 3c), but it is dependent on winter mean values.”

Revised text:

“Mean spring $\Delta f\text{CO}_2$ also has a significant relationship to annual means ($R^2 = 0.82$; Fig. 3c), because the seawater $f\text{CO}_2$ levels at the end of winter determine the starting point from which springtime processes, such as primary productivity and meltwater dilution, reduce seawater $f\text{CO}_2$.”

1.7. Figure 4: check the years in the legend for panel b, should 2022 be 2020?

Thanks for pointing this out; we have checked and the years are correct. Our dataset for most of the marine variables go until 2020, but our sea ice observations dataset runs until 2022. The intention is to show that patterns seen between high sea ice/high stratification and low sea ice/low stratification years apply for the entire time series for which we have data. We have now clarified this in the caption of Figure 4.

1.8. Lines 154-159: did you verify the dominant modes of variability (ENSO, SAM)? A mention on this is relevant for the study area, as the high stratification years appear to be years of ENSO+, for example.

Thank you for bringing up the topic of ENSO and SAM, as it is indeed relevant to the study area and to the stratification of the water column. We had initially included it in the discussion, but removed it for the final version for reasons of conciseness and focus on the main messages. However, we recognise the importance of bringing up this topic. Instead of including it around lines 154-159, we have opted to bring it up at the end of the discussion (i.e. before the Conclusions), because we felt we could explain the relevance to ocean CO_2 uptake more clearly once the mechanisms had been explained. It also linked well with addressing comment #1.9. Please see the revised text where we address comment #1.9.

1.9. Lines 217-227: Great to see the caveats indicated here. Perhaps also report something related to the verified study area (Ryder Bay), since this is a very localized investigation to be representative of large coastal areas of the Southern Ocean.

We agree with this suggestion. We have added the following text to the end of the main text, i.e. before the Conclusions. Please note that this text also addresses comment #1.8.

*“Given the important role of sea ice on the interannual variability of sea-air CO_2 flux, we would here like to highlight a link to modes of large-scale atmospheric variability, such as the Southern Annular Mode (SAM) and the El Niño Southern Oscillation phenomenon. The former is broadly characterised by an oscillation in surface atmospheric pressure between a node over Antarctica and a ring over lower latitudes, with consequent fluctuations in the dominant westerly winds overlying the Southern Ocean, whilst the latter is focussed on variability emanating from the equatorial Pacific Ocean, but with high-latitude impacts (**Thompson and Wallace, 2000; Turner, 2004**). A positive SAM occurred in 2015, followed by an El Niño event. The pairing of these two phenomena caused anomalous atmospheric cooling in winter along the WAP (**Clem et al., 2016**), resulting in prolonged sea ice cover in*

both 2015 and 2016. According to the mechanisms discussed in the current study, this explains higher wintertime stratification, lower mean winter $\Delta f\text{CO}_2$, and higher annual atmospheric CO_2 uptake in Ryder Bay compared to years with less sea ice cover. Modes of atmospheric variability clearly play an important role in the interannual variability of sea-air CO_2 flux along the WAP. Their key role for open ocean carbon uptake has further been demonstrated by other studies (e.g., **Hauck et al., 2013; Lenton et al., 2007; Lovenduski et al., 2007; Santos-Andrade et al., 2023**), indicating that wind intensity associated to climate modes regulates wintertime deep-water ventilation and therefore the interannual variability of the Southern Ocean CO_2 sink (**Mayot et al., 2023**). Given that the conditions of Ryder Bay are inherently local, they cannot be taken as indicative of the broader WAP or Southern Ocean (**Jones et al., 2017, Venables and Meredith, 2014**). However, by using these locally-collected, year-round observations, we have been able to explore the direct mechanisms behind ocean CO_2 uptake variability, including the role of sea ice and wintertime processes, which remain largely unexplored for other parts of the Southern Ocean, especially within seasonally sea ice-covered regions.”

1.10. Lines 250-254: This paragraph needs to be better elaborated, mainly to highlight the contribution that previous studies (e.g. 12, 30) have already made to the topic of the impact of sea ice on CO_2 flux estimates. An additional reference may also help in this discussion: <https://doi.org/10.1029/2022GB007518>, regarding the marine biogeochemistry close to the study area.

We agree with this comment. We have revised the text, and in doing so made a few alterations in the lines following lines 250-254. Thank you for pointing us towards the manuscript by Santos-Andrade et al. We decided not to cite it in this particular part of the manuscript, because we want to keep it focused on the role of wintertime processes on seasonal/interannual variability, which is not the focus of the study by Santos-Andrade et al. Instead, we have included a citation of it where we briefly discuss SAM and ENSO (see our responses to comments #1.8 and #1.9).

Original text:

“Our unique case study on Ryder Bay highlights important wintertime mechanisms that need to be considered in similar Antarctic coastal regions, which contribute significantly to the Southern Ocean CO_2 flux [3, 12, 18, 29, 30]. Substantial seasonal and interannual variability in their marine carbonate systems has been observed, but not yet fully understood [30–32].

Although Ryder Bay is not representative of open ocean conditions, the mechanisms presented here should be tested on larger scales in the Southern Ocean where winter data is chronically lacking. We have shown that processes in winter can be more important than processes in summer to explain interannual variability of the marine carbonate system. Assessments of net CO_2 exchange based predominantly on summer data might therefore not resolve critical processes, or adequately resolve the magnitude, of the interannual variability of Southern Ocean CO_2 uptake.”

Revised text:

“Previous studies have explored how sea ice shapes the seasonal variability of sea-air CO_2 flux by dampening wintertime CO_2 outgassing and enhancing summertime

CO₂ uptake, particularly for coastal regions (**Monteiro et al., 2020, Legge et al., 2017**). Substantial interannual variability has also been observed in previous work (**Lencina-Avila et al., 2018; Monteiro et al., 2020; Droste et al., 2022**), but how this variability is affected by sea ice is not yet fully understood. To develop a better understanding, our unique case study on Ryder Bay suggests that ice-ocean feedbacks on ocean-interior processes need to be considered, especially in winter. These findings complement similar conclusions for interannual variability in ocean stratification (**Venables & Meredith, 2014**), phytoplankton blooms and biomass (**Clarke et al., 2008; Rozema et al., 2017**), phytoplankton distribution in the water column (**Venables et al., 2013**), community composition (**Rozema et al., 2017; Saba et al., 2014**), and nutrient cycling (**Henley et al., 2017**).

The coastal regions of Antarctica, such as Ryder Bay, collectively constitute a significant contribution to the overall Southern Ocean CO₂ flux (**Arrigo et al., 2008b; Roden et al., 2013; Legge et al., 2015; Monteiro et al., 2020**). Hence, it is important to understand the key mechanisms and feedback processes operating here. Whilst open ocean conditions across the broad Southern Ocean are necessarily different, identification of important mechanisms in such coastal regions can inform future research priorities more widely. In particular, our results reveal the importance of wintertime processes in explaining interannual variability in the marine carbonate system, which is supported by open Southern Ocean studies that suggest strong links between wintertime stratification and carbon cycling (**Giddy et al., 2023**). Assessments of net CO₂ exchange based predominantly on summer data are therefore unlikely to resolve critical processes affecting the interannual variability of Southern Ocean CO₂ uptake. Our findings emphasise a need to expand and sustain year-round observations of dynamic ice-ocean interactions in seasonally sea ice-covered regions to better capture their role in a changing polar environment.”

1.11. Lines 257-259: This sentence should be attenuated, indicating other processes that may not have been captured by the data evaluated in this study.

We have revised this sentence (see below). Please note that we have also revised the entire paragraph in response to comment #1.10.

Original text:

“We have shown that processes in winter can be more important than processes in summer to explain interannual variability of the marine carbonate system.”

Revised text:

*“In particular, our results reveal the importance of wintertime processes in explaining interannual variability in the marine carbonate system, which is supported by open Southern Ocean studies that suggest strong links between wintertime stratification and carbon cycling (**Giddy et al., 2023**).”*

1.12. Line 314: include the value of the propagated error to obtain the estimate of CO₂ fugacity.

We have now included the propagated uncertainty from the uncertainties of DIC and TA, for fCO₂, by adding the following sentence on line 314:

“The mean propagated uncertainty for $f\text{CO}_2$, from the analytical uncertainties of DIC and TA, is 8 μatm , as calculated by PyCO2SYS [38].”

1.13. Lines 335 and 341: do not need to abbreviate “asl”.
Done.

1.14. Lines 342-451: it is not clear how this scaling was performed. I suggest including more details and information, since the indicated reference (3) also does not make it clear how this adjustment is included in equation 1.

Thank you for this point of feedback. We have now revised the text (see below). We also edited Eq 1 (Methods section *$f\text{CO}_2$ and CO_2 flux calculations*), which now explicitly includes the linear scaling we explain in the text. We have also re-ordered the factors in the equation to be consistent with the order in which they are discussed in the text.

Original text:

“Sea-air CO_2 flux is attenuated by the presence of sea ice, which we account for by applying a simple linear scaling to the gas transfer velocity according to the fraction of open water [3]. At 100 % sea ice cover, the gas transfer velocity is scaled down by a maximum of 99 % (i.e., always leaving a minimum of 1 % “open water”) [3, 18, 52, 53]. This is done in an attempt to include a representation of the complex mechanisms that explain sea ice-atmosphere gas fluxes recorded in previous studies [54–57]. However, this has negligible effect on CO_2 flux compared to when we assume zero flux at 100 % sea ice cover, ranging between -0.2 and $0.2 \text{ mol m}^{-2} \text{ yr}^{-1}$. We use sea ice cover of all types when scaling the gas transfer velocity, unless otherwise specified.”

Revised text:

“Sea-air CO_2 flux is strongly attenuated by the presence of sea ice, which we account for by linearly scaling the gas transfer velocity, k , with the fraction of open water, determined in Eq. 1 by $(1 - f_{\text{sic}})$, where f_{sic} represents the fraction of sea ice cover (3). We use sea ice cover of all ice types when scaling k , unless otherwise specified. Following Legge et al., at 100 % sea ice cover we set f_{sic} to 0.99 instead of 1. This means that k is only ever scaled down by a maximum of 99 %, allowing for a small amount of CO_2 flux to occur at all times [54–57]. This minimum flux represents the possibility for sea-air gas exchange through cracks in the ice, as well as sea ice-air gas exchange observed in previous studies [54–57]. However, the effect of this k attenuation limit on total CO_2 flux is negligible ($< 0.2 \text{ mol m}^{-2} \text{ yr}^{-1}$), likely because $\Delta f\text{CO}_2$ is relatively low when sea ice cover is 100 %.”

Congratulations on the work! It was a very pleasant read.

Thank you ☺

Reviewer #2 (Remarks to the Author):

Summary

The manuscript by Droste et al. is well-written and provides in-depth discussions. The study presents 10 years of surface water carbonate chemistry data and CTD profiles at the Rothera Time Series on the West Antarctic Peninsula (WAP), including continuous winter observations - a rare dataset for coastal Antarctic. The main finding highlights the significant influence of sea ice condition and water stratification in winter/spring on interannual variability in air-sea CO₂ fluxes. Specifically, reduced sea ice coverage and lower stratification in winter lead to increased CO₂ outgassing during the winter to spring transition, and a 20% reduction in annual carbon drawdown. This is an important and timely research topic, and I believe it would be of great interest to the research community as well as broader audiences for Comm. Earth & Env. interested in climate change and carbon cycling. While the study has explored the physical processes, **biological processes (photosynthesis and respiration) are major drivers of annual carbon flux in this region and should be further discussed and incorporated in the proposed mechanism. For instance, could under-ice or ice-edge phytoplankton growth contribute to the spring CO₂ drawdown observed in high stratified years? could under-ice or ice-edge phytoplankton growth contribute to the spring CO₂ drawdown observed in high stratified years? (Hague and Vichi 2021, Biogeosciences, 18:25-38). In low stratified years, would deeper winter mixing (resulting in higher nutrient availability) and a longer open water period enhance biological carbon drawdown when integrated over the growing season?**

- 2.1. Discussion of biological processes as drivers in annual carbon flux. This feedback point has been taken from the summary text of Reviewer #2, **marked in bold.**

Thank you for bringing up this interesting topic. We agree that biological processes play an extremely important role in annual CO₂ uptake. While they are certainly drivers of the total CO₂ uptake, the data in our work suggests that it is the wintertime processes (i.e. physical processes in winter) that regulate the interannual variability in mean annual CO₂ uptake, rather than summertime processes, including biological CO₂ uptake (**we have clarified this in the revised text, below**). We would like to clarify two points:

1. Spring and summertime exhibit substantial year-to-year variability (see mean values for $\Delta f\text{CO}_2$ and CO₂ flux in Fig. 3), but it does not seem to govern the interannual variability of mean $\Delta f\text{CO}_2$ and CO₂ flux (at least for Ryder Bay). Instead, it is the wintertime $\Delta f\text{CO}_2$ and CO₂ flux that is correlated to annual means. On a larger scale, this is consistent with **Mayot et al. (2023)** who found that physical processes in wintertime (i.e. ventilation of deep water masses) governed the interannual variability of the Southern Ocean CO₂ sink.
2. There are indeed inter-seasonal dependencies, including dependencies of spring and summertime processes, including biological processes on wintertime processes, i.e. stratification. Wintertime stratification has an effect on processes in subsequent seasons, including stratification (**Venables & Meredith, 2014**), phytoplankton blooms and biomass (**Clarke et al., 2008; Rozema et al., 2017**), phytoplankton distribution in the water column (**Venables et al., 2013**), community composition (**Rozema et al., 2017**),

ecological community composition (**Saba et al., 2014**) and nutrient cycling (**Henley et al., 2017**). Since sea ice microalgal species also have seeding potential (**van Leeuwe et al., 2022**), it would be interesting to investigate differences in pelagic phytoplankton blooms between years with earlier and later sea ice retreat. Overall, high- and low-stratification winters matter to subsequent spring and summers, and will therefore also affect spring/summertime seawater $f\text{CO}_2$. However, we are unable to disentangle and quantify the contribution of these various processes on $f\text{CO}_2$ with our dataset. Moreover, the net effect of all these complex processes does not seem to govern the interannual variability observed in the annual $\Delta f\text{CO}_2$ and CO_2 flux.

Based on the explanation given above, we decided not to incorporate biological processes in the mechanism we propose. However, we agree that clarification on this topic, as well as an acknowledgement to biological processes and inter-seasonal dependencies will improve the manuscript.

Therefore, to clarify point 1, we have revised the text on lines 135-137, where we think this clarification is most impactful:

Original text:

“We derive a similar conclusion for the CO_2 flux; winter means explain most of the interannual variability (Fig. 3e-h).”

Revised text:

*“We derive a similar conclusion for the CO_2 flux (Fig. 3e-h). Thus, winter means explain most of the interannual variability, despite substantial year-to-year variability in mean summertime values and the important role of summertime processes in net atmospheric CO_2 uptake. On a larger scale, this is consistent with findings from **Mayot et al. (2023)**, who found that deep water mass ventilation in winter governs interannual variability of the Southern Ocean CO_2 sink.”*

To clarify point 2, we have added the following text on line 217:

Added text:

*“While our findings highlight the relevance of physical wintertime processes on interannual variability of CO_2 uptake, we want to remark that other studies showed effects of wintertime stratification on both physical (**Venables & Meredith, 2014; Venables et al., 2013**) and biogeochemical processes (**Clarke et al., 2008; Rozema et al., 2017; Saba et al., 2014; Henley et al., 2017**) in the following spring and summer. The net effect of these processes on marine carbonate chemistry during the productive season is likely complex. Whilst we are unable to explore these effects here, they are notable topics for future research.”*

Detailed comments

- 2.2. Line 53-54: Could you clarify the limitations here? Are they due to calibration issue or related to parameters that can only be measured through ship-based observations?

Limitations of the float-based CO₂ uptake estimates are due to uncertainties associated to the pH sensor mounted on biogeochemical floats. These sensors are prone to drift, which is tricky to correct. We have clarified this in the text, according to the following revision.

Original text:

“While Biogeochemical Argo floats are closing the winter data gap (Bushinsky et al., 2019), float-based estimates on winter CO₂ outgassing have unresolved uncertainties due to insufficient ship-based in-situ observations for evaluation of winter processes.”

Revised text:

*“While biogeochemical Argo floats are closing the winter data gap using pH sensor-based measurements (Bushinsky et al., 2019), float-based estimates of winter CO₂ flux have unresolved uncertainties caused by sensor drift, measurement biases (**Bushinsky et al., 2025**), and insufficient ship-based in-situ observations for evaluation of winter processes.”*

2.3. Line 121: linear scaling.

Done.

2.4. Line 286: specify the approximate depth range for the CTD casts.

Done. Text has been edited accordingly, and now reads as:

“CTD casts profiled the top 200 m of the water column until 2003, after which full depth casts were deployed, i.e. down to 500 m for Site 1, to 300 m for Site 2, and to 100 m for Site 3 (Venables et al., 2023).”

2.5. Line 374 – 37: While already cited Legge et al., it would be helpful to list the end member values (i.e., salinity and delta 18O) used in the calculation.

Done. Ideally, we would have liked to provide this information in the format of a table, but we are at capacity when it comes to figures/tables, according to the formatting guidelines for this journal. The text therefore now reads:

“For consistency, we use the same end-members as Legge et al. [12]: salinity end-members are 34.62, 7, and 0 for mCDW, sea ice meltwater, and meteoric water, respectively; and $\delta^{18}\text{O}$ end-members are 0.08 ‰, 2.1 ‰, and -17 ‰ for mCDW, sea ice meltwater, and meteoric water, respectively.”